Human population and socioeconomic modulators of conservation performance in 788 Amazonian and Atlantic Forest reserves

de Marques Ana Alice B. anaalice@cpovo.net 1 2
Schneider Mauricio 2 3
Peres Carlos A. 2
1 Assessoria Legislativa, Câmara Legislativa do Distrito Federal , Brasília , Distrito Federal , Brazil
2 School Environmental Sciences, University of East Anglia , Norwich , Norfolk , United Kingdom
3 Consultoria Legislativa, Câmara dos Deputados , Brasília , Distrito Federal , Brazil
Xu Jianhua
Electronic publication date: 2016 Jul 14
Publication date: 2016
Volume: 4
Electronic Location ID: e2206
Received 2015 Dec 5; Accepted 2016 Jun 13
Copyright: ©2016 De Marques et al.
Copyright year: 2016
Copyright holder: De Marques et al.
License: This is an open access article distributed under the terms of the Creative Commons Attribution License, which permits unrestricted use, distribution, reproduction and adaptation in any medium and for any purpose provided that it is properly attributed. For attribution, the original author(s), title, publication source (PeerJ) and either DOI or URL of the article must be cited.
License URL: https://creativecommons.org/licenses/by/4.0/

Keywords: Brazil, Protected areas, Forest biomes, Atlantic Forest, Amazon, Conservation performance, Human population density

Funding: Brazilian Ministry of Education (CAPES) 004/2012 CAP was supported by a research collaboration grant (004/2012) from the Brazilian Ministry of Education (CAPES). The funders had no role in study design, data collection and analysis, decision to publish, or preparation of the manuscript.

==============================
Protected areas form a quintessential component of the global strategy to perpetuate tropical biodiversity within relatively undisturbed wildlands, but they are becoming increasingly isolated by rapid agricultural encroachment. Here we consider a network of 788 forest protected areas (PAs) in the world’s largest tropical country to examine the degree to which they remain intact, and their responses to multiple biophysical and socioeconomic variables potentially affecting natural habitat loss under varying contexts of rural development. PAs within the complex Brazilian National System of Conservation Units (SNUC) are broken down into two main classes—strictly protected and sustainable use. Collectively, these account for 22.6% of the forest biomes within Brazil’s national territory, primarily within the Amazon and the Atlantic Forest, but are widely variable in size, ecoregional representation, management strategy, and the degree to which they are threatened by human activities both within and outside reserve boundaries. In particular, we examine the variation in habitat conversion rates in both strictly protected and sustainable use reserves as a function of the internal and external human population density, and levels of land-use revenue in adjacent human-dominated landscapes. Our results show that PAs surrounded by heavily settled agro-pastoral landscapes face much greater challenges in retaining their natural vegetation, and that strictly protected areas are considerably less degraded than sustainable use reserves, which can rival levels of habitat degradation within adjacent 10-km buffer areas outside.

Introduction

Protected areas worldwide are essential for the conservation of biological diversity. The global network of protected areas (PAs) has increased exponentially over the last two decades, especially in tropical regions (Jenkins & Joppa, 2009). The World Database on Protected Areas (WDPA) records some 178,000 terrestrial PAs under different protection categories (IUCN & UNEP-WCMC, 2015), which globally account for 12.7% of the land area, including inland water bodies. However, although the total number of PAs has increased by about 58% since 1990, overall coverage is far from uniformly distributed across ecoregions (Schmitt et al., 2009). Furthermore, the effective level of protection within a PA also depends on its management category, which in tropical countries is highly skewed towards multiple-use sustainable development reserves where use restrictions are more relaxed (Peres, 2011).

Only 7.7% of all the PAs in forest biomes worldwide are strictly protected according to the International Union for Conservation of Nature (IUCN categories I–IV), whereas 13.5% are in less restrictive management categories (IUCN categories V and VI) (Schmitt et al., 2009). This poses additional concerns about the effectiveness of multiple use and extractive PAs in terms of maintaining ecosystem integrity and preventing biodiversity loss. To what degree, therefore, is the long-term integrity of formally designated PAs determined by their size and management category? This is a recurrent question in several studies evaluating the extent to which different ecoregions are represented by PA networks (Chape et al., 2005; Jenkins & Joppa, 2009). However, few studies have considered the concomitant roles of both size and management category as measurable indicators of reserve conservation performance (Joppa, Loarie & Pimm, 2008; Peres, 2011; Miranda et al., 2016). More often researchers have focused on the patterns of land-use and human economic activities within and around each PA (DeFries, Karanth & Pareeth, 2010; Beresford et al., 2013; Pfaff et al., 2015; Bowei et al., 2016).

Throughout the tropics, relentless human population growth, and expanding agricultural frontiers, infrastructure projects and industrial development have fostered rapid primary habitat conversion, both increasing pressure on existing PAs and reducing opportunities to create new reserves. Tropical forest regions, in particular, have undergone rapid changes in land-use intensification, leading to increasing isolation and habitat degradation of existing PAs (DeFries et al., 2005; Wright, 2005). These mounting pressures combined with conflicts with powerful economic interests have also led to formal alterations in existing environmental legislation, ultimately resulting in the downsizing, downgrading, and even degazettement of many formally established PAs (Mascia & Pailler, 2011; Marques & Peres, 2015).

Brazil is the largest tropical country on Earth, and contains some 41% of world’s remaining tropical forests and approximately 13% of all known species (Lewinsohn & Prado, 2005). Between 2000 and 2005, however, Brazil lost an average of ∼33,000 km2 of forest each year, the fastest absolute tropical deforestation rate in human history (Hansen, Stehman & Potapov, 2010). Credible projections suggest that primary habitat conversion will continue to increase as the country becomes one of the largest emergent agricultural and industrial economies, amounting to a rapid escalation in demand for new arable cropland, energy, and raw materials. To boost economic growth, the Brazilian government has launched an ambitious macroeconomic development blueprint—the Growth Acceleration Plan (PAC)—which envisages to deliver many mega infrastructure projects, including major hydroelectric dams, power transmission lines, and highways and waterways, to hitherto poorly accessible ‘hinterland’ regions. Such concerted geopolitical strategies will clearly have a major impact on natural ecosystems, especially in remote parts of Amazonia.

Brazil is also the only country hosting two of the world’s major tropical forest biomes, which are disjunct across a wide latitudinal gradient spanning the central-northern (Amazonia) and eastern-southern regions of the country (Atlantic Forest). These biomes experienced very different post-colonial histories of European conquest since 1500, setting up a unique context of marked socioeconomic polarity in regional scale historical development. While the Atlantic Forest region experienced a long history of logging and forest conversion into agriculture since the early 16th century, Amazonia remained roadless and virtually isolated from the rest to the country until 1970.

The Atlantic Forest domain straddles along the entire eastern coast and inland continental areas farther south across over 23 degrees in latitude, was settled by European colonists since the early 1500s, and is currently occupied by ∼70% of the entire Brazilian population of ∼205 million. The Atlantic Forest domain is highly heterogeneous and includes coastal and montane evergreen forests, semideciduous seasonally-dry forests, dunes, marshes along coastal plains, and native grasslands, all of which amount to a global biodiversity hotspot containing 19,355 plant species, ∼40% of which endemic (Forzza et al., 2012). Of the original Atlantic Forest vegetation cover of 1.3 million km2 (13% of Brazil’s territory), only 22.2% remains, considering all ecoregions and centers of endemism (official estimates from IBAMA, 2012). Unlike the vast forest tracts of Amazonia, forest remnants in the Atlantic Forest are now highly fragmented and largely restricted to existing reserves, which protect ∼9% of the remaining vegetation cover (Ribeiro et al., 2009; Tabarelli et al., 2010). Growing demographic and economic pressures over five centuries have rendered the Atlantic Forest into one of the most threatened biodiversity hotspots worldwide (Fundação SOS Mata Atlântica & INPE, 2013).

In contrast, Brazilian Amazonia hosts the largest unbroken tract of tropical forest controlled by a single country (∼4.2 million km2), and one of the highest levels of known and unknown biological diversity anywhere (Peres, 2005; Pimm et al., 2014). The region accounts for some 55% of Brazil’s territory but contains only about 11.8% of the Brazilian population. In addition to the largest surface and underground freshwater reserves, this region contains one of the largest untapped mineral reserves on Earth, as well as vast areas of cheap agricultural land and a flat relief that facilitates mechanized farming and cattle ranching. The historical context is also in marked contrast to the Atlantic Forest, as Brazilian Amazonia remained entirely roadless and unexploited by agropastoral interests until 1970, when only <1% of the region had been deforested and the first major paved highway linked eastern Amazonia to the rest of Brazil (Peres et al., 2010).

Brazil has established a wide network of protected areas, including formal nature reserves (referred to as ‘conservation units’), indigenous lands, and quilombola territories (traditional Afro-Brazilian communal territories). However, conservation units are the only officially recognized reserves that have a strong legal basis in Brazil to ensure biodiversity protection. The formal regulations for creating and managing Brazilian conservation units were sanctioned by Law 9.985/2000, which also established the National System of Nature Conservation Units (Portuguese acronym: SNUC). Brazilian parks and nature reserves are classed under 12 management categories with varying degrees of protection, ranging across the entire spectrum of reserve types (IUCN categories I–VI). We therefore restricted our analysis to PAs under SNUC regulations, comprising all formally established conservation units within Amazonia and the Atlantic Forest. Unlike protected forests in Europe and North America, where IUCN categories V and VI are most frequently adopted (Schmitt et al., 2009), these two major tropical forest biomes host a considerable number of PAs under all management categories.

Detailed georeferenced data on species richness and diversity are only available at selected sites, either within or outside reserves, and most studies use deforestation (or the lack thereof) as the proxy of protected area performance in preserving forest biodiversity (e.g., Nepstad et al., 2006; Beresford et al., 2013; Paiva, Brites & Machado, 2015; Pfaff et al., 2015; Ren et al., 2015; Bowei et al., 2016; Miranda et al., 2016). Here, we provide a quantitative assessment of the degree to which the natural vegetation cover of 788 Brazilian forest reserves within the Amazonian and the Atlantic Forest domains have been converted into different patterns of land use. In particular, we examine the role of reserve management category (under the jurisdiction of either federal or state-level management agencies) on this metric of conservation performance both within and around reserve boundaries. Unlike other studies addressing a small number of reserves or presenting aggregate deforestation data from areas within and outside PAs, we relate the absolute and relative amounts of vegetation conversion within reserve polygons to areas immediately outside. To test the assumption that human dwellings and economic activities have a negative impact on PAs, and that PA category determines the degree to which PAs succumb to such adverse effects, we explicitly consider the effects of reserve size, human population density, and per capita wealth within and around each reserve on measures of reserve performance. Finally, we consider how human population density, socioeconomic context and conservation investment capacity as both part of the problem and the solution governing the fate of tropical forest reserves.

Methods

Protected area categories

We examined all forest protected areas encompassed by both the Brazilian Amazon and the Brazilian Atlantic Forest biomes, which had been legally sanctioned under the SNUC legislation. Official PAs within the SNUC comprise federal, state and municipal county conservation units and are divided into two main management classes: (1) strictly protected reserves, with five categories; and (2) sustainable use reserves, with seven categories. The first group essentially aims to target conservation objectives, and are restricted to non-consumptive natural resource use (equivalent to IUCN categories I–IV). The second group legally recognizes human occupation and sustainable use of natural resources, typically allowing human occupation of the areas, including agricultural activities (IUCN categories IV–VI). A reserve under IUCN category IV may be either strictly protected or sustainable use, depending on the nominal SNUC category it belongs to (but in this analysis we grouped all PAs according to their legal restrictions within Brazil). For analytical purposes, we grouped all reserves into eight categories, in terms of their overall group, level of protection, land-use restrictions, and conservation objectives (Table 1). We excluded from the analysis other types of PAs that may also afford legal protection, such as indigenous lands, quilombolas, and private areas of restricted use, but are not governed under the SNUC legislation.

Table 1 Categories of Amazonian and Atlantic Forest protected areas considered in this study, followed by a description of key management restrictions and equivalent international status according to the International Union for Conservation of Nature.

SNUC reserve category	IUCN category	PA acronym	Description	
Strictly protected reserves	
Biological reserve	IUCN Ia	Reserve	Categories including the most restricted use, comprised of public areas aiming to preserve intact ecosystems. No resident human population are permitted.	
Ecological station	IUCN Ib	
Park	IUCN II	Park	In the public domain, aiming to preserve areas of great ecological importance and scenic beauty. Only research, recreation and tourism activities are permitted.	
Natural monument	IUCN III	Refuge	A Natural Monument contains areas of outstanding natural beauty, whereas a Wildlife Refuge aims to protect areas in which species can persist and reproduce. Both categories can include private lands, as long as land use is compatible with the objectives of each category. Otherwise, landholdings can be expropriated and incorporated into the public domain.	
Wildlife refuge	IUCN IV	
Sustainable use reserves	
Environmental protection area	IUCN V	EPA	Aims to protect biological diversity, but allows human occupation and natural resource use/extraction. Consists of public and private lands, and often includes agricultural and/or urban areas.	
Area of relevant ecological interest	IUCN IV	AREI	Usually small sites with little or no human occupation. Contains important examples of biota and their use can be carefully regulated. Can include either public or private lands.	
Forest	IUCN VI	Forest	Predominantly native forest cover whose main objective is the sustainable use of natural resources. Under public domain, with public visitation permitted.	
Extractive reserve	IUCN VI	SUST	All categories comprise areas in the public domain that mainly aim to manage natural resources. Both Extractive Reserves and Sustainable Development Reserves contain traditional populations conducting extractive livelihoods.	
Fauna reserve	
Sustainable development reserve	
Natural heritage private reserve	IUCN IV	NHPR	A private landholding, pledged in perpetuity for the conservation of biological diversity. Research, ecotourism visitation, recreation and education activities are all permitted.	

Data acquisition and geoprocessing

Our study regions cover the two largest forest biomes in tropical South America, Amazonia and the Atlantic Forest, the phytogeographic boundaries of which are defined by the Brazilian Institute of Geography and Statistics (IBGE). Shapefiles describing the geographic boundaries of all conservation units were obtained from official sources (mapas.mma.gov.br and www.icmbio.gov.br/portal/comunicacao/downloads.html). Complementary information about each protected area was extracted from the Brazilian National Registry of Conservation Units (CNUC—www.mma.gov.br/areas-protegidas/cadastro-nacional-de-ucs). In total, we consider reserve polygons of 788 federal, state and municipal county scale conservation units that met all of the following criteria: reserve boundaries included natural forest cover and overlapped one of these two forest biomes; satisfactory correspondence in each reserve number code between its shapefile and that of the CNUC database; conservation units were terrestrial rather than marine reserves; and reserves were represented by a reliable polygon comprising an area of at least 2 ha (many small private forest reserves were represented by small circles in the shapefile, due to lack of accurate mapping, so were excluded from the analysis). Our overall sample corresponds to 82% and 62% of all Amazonian and Atlantic Forest conservation units, respectively (MMA, 2012). In the latter biome, most private reserves contained in shapefiles did not meet our size criteria nor data quality control in terms of their spatial data and were therefore also excluded from the analysis.

There is considerable spatial overlap between conservation units in Brazil, partly because strictly protected conservation units can be located within sustainable use conservation units (e.g., many Environmental Protection Areas (EPAs) may include parks, ecological stations, or adjacent private areas). There are also mapping errors and competition between federal, state and municipal level environmental agencies, which may set aside overlaying PAs within existing reserves, thereby claiming jurisdiction over their respective territories. Therefore, to avoid overestimating reserve areas and their respective classes of land cover, these overlaps were painstakingly manually removed from the vector files (Fig. S1). The following hierarchical structure was used to decide which conservation unit should prevail in cases of overlapping areas: (1) legal restrictions on land use (e.g., strictly protected reserves prevailed over sustainable use reserves, ecological stations over parks, and extractive reserve over EPAs); (2) official year of decree (oldest reserves prevailed); (3) reserve boundaries completely enclosed within another conservation unit (if an EPA or Natural Monument overlapped a smaller private reserve, the latter was retained and the conservation unit larger than the overlap zone was subtracted of a corresponding area).

To compare each conservation unit with its surrounding landscapes, a 10-km external buffer was created from the reserve perimeter, and any overlap between the 788 buffers and neighboring conservation units were also removed. A current land cover map was then generated overlaying the deforestation data (see below) to the vegetation map of Brazil (2002), at a scale of 1:250,000, obtained from the Ministry of the Environment (mapas.mma.gov.br). For Amazonia, this overlay used deforestation polygons up to the year 2011, at a scale of 1:250,000, from the Deforestation Monitoring Project for the political region of Legal Amazonia (PRODES) obtained from INPE (www.obt.inpe.br/prodes). For the Atlantic Forest biome, deforestation data up to 2008 were obtained at a scale of 1:50,000, from the Satellite Deforestation Monitoring Project for Brazilian Biomes (PMDBBS) obtained from IBAMA (siscom.ibama.gov.br/monitorabiomas). The vector files from these three sources were overlayed onto a raster map with a resolution of 1 millidegree (∼109 m), which is compatible with a 1:250,000 scale of analysis.

Human population density (HPD) was calculated considering data from the last national census of the Brazilian population (2010), from which a shapefile of 213,872 sub-municipal census district polygons (also known as ‘census sectors’) covering the two major forest biomes (Amazonia: 18,031; Atlantic Forest: 195,841) was generated (www.ibge.gov.br). Municipal county boundaries were then linked to the county-level gross domestic product (GDP) dataset (www.ibge.gov.br) to estimate the county-scale per capita GDP as of 2009. The same boundaries were also linked to the Human Development Index (HDI) at the municipal scale (www.pnud.org.br). HPD and GDP, which were represented at a census district and municipal scale, respectively, were estimated for each conservation unit and its corresponding buffer zone, on the basis of the area-weighted average by intersecting the PA polygon with either the census districts or municipal counties. All area calculations were carried out using the Albers Equal Area Conic projection, South American 1969 datum.

Data analysis

Generalized linear models (GLMs) were used to investigate predictors of natural forest loss within all forest reserves of different denominations in each biome and both biomes combined. Our response variable was the cumulative conversion rate of any natural vegetation within each of the 788 mapped protected areas across our bi-regional sample. Our predictors included several key variables describing the reserve size, reserve age (year of decree), reserve category, management class (strictly protected or sustainable use reserve), weighed mean human population density (HPD) at the scale of census district both within each reserve polygon and the buffer zone neighboring this polygon (calculated on the basis of all terrestrial areas only), two socioeconomic variables (weighed mean Gross Domestic Product (GDP) and Human Development Index (HDI) at the scale of municipal counties), and reserve governance structure (at the level of federal, state-level or municipal administration). HPD (log10 x + 1) within and outside reserves was highly correlated (r = 0.789), so we used either one rather than both of these variables in any given model. In comparison, GDP and HDI within and outside reserves were less strongly correlated (r = 0.298–0.331) and could be considered as independent from one another. We then tested for multicollinearity among variables by examining the least moderately redundant or collinear Variation Inflation Factors, but no variables were sufficiently collinear at a VIF ≥ 5 threshold (Dormann et al., 2013). The relative strength of these predictors was then examined using multiple GLMs to understand their role as drivers of forest conversion rates. Rather than treating rates of forest loss as proportional data which has a number of drawbacks (Warton & Hui, 2011), we explicitly considered the total extent of primary habitat loss by modelling the total area (ha) of forest conversion within each reserve, but used the total terrestrial reserve size (ha) as an offset variable, and a quasipoisson error structure to avoid overdispersion. We then repeated this modelling approach using a failure:success binomial error by considering the total number of hectares that were either converted to other land-uses or retained in apparently intact form within each reserve. Models were examined on the basis of minimum BIC and AICc values, and there was good convergence in identifying the most parsimonious “best” model. These analyses were performed with data from both biomes combined, as well as separately. All models were fitted using the ‘lme4’ package (Bates et al., 2007) within the R platform. To assess the relative strength of spatial effects on the characteristics of forest reserves across the two biomes, we used spatial multiple linear regression implemented with simultaneous autoregression using the spautolm function in the R package ‘spdep’. The degree of within-reserve forest loss was clearly structured in space across all Amazonian and the Atlantic Forest protected areas, and the autoregressive parameter λ indicated significant spatial autocorrelation across all reserves (β = 0.72, p < 1e–15). However, this successfully eliminated spatial autocorrelation of the residuals (Moran’s I, p = 0.64). Finally, we used paired t-tests to examine differences in deforestation rates and HPD within and outside protected areas.

Results

Of the 788 forest reserves considered here, which encompassed a total area of 120,289,994 ha, 251 are distributed across the Amazon (111,334,941 ha) and 537 across the Atlantic Forest (8,955,053 ha) (Fig. 1 and Fig. S2). In general Atlantic Forest reserves share an older history since they were first established (see Fig. S3), but account for only ∼8.0% of the total area of Amazonian reserves. Sustainable use reserves (hereafter, SURs) accounted for 64.5% (508) of all forest reserves and 63.9% of the total area of the reserves we considered, whereas strictly protected reserves (hereafter, SPRs) accounted for 35.5% (280) of the reserves and 36.2% of the total area (Table 2). The overall proportion of the original natural vegetation (in almost all cases primary forest) converted within those reserves (Amazonia: 12.1%; Atlantic Forest: 44.5%) was lower than that within the surrounding buffer areas (Amazonia: 53.1%; Atlantic Forest: 65.7%) for all reserve categories (paired t-test, t = 12.23, p < 0.0001). However, there were many exceptions for which vegetation conversion rates within reserves (33.9 ± 30.2%, N = 537) were actually greater than those outside (62.1 ± 24.7%), particularly for SURs in the Atlantic Forest. These trends in land use change reflect marked regional differences in human population densities, which were much higher both within and outside Atlantic Forest reserves than those within and outside Amazonian reserves (Fig. 2).

Figure 1 Land cover across the Brazilian Amazon and the Brazilian Atlantic Forest biomes, and their respective 788 forest reserves.

Overall boundaries of the two biomes are shown in the small inset map. Forest cover is indicated by green areas; deforested areas converted to other land-uses are shown in light pink. Strictly protected and sustainable use reserves are delineated by red and gray lines, respectively.

Table 2 Area and percentage of forest loss both inside and outside the 788 federal, state and municipal county Conservation Units in the Amazon and Atlantic Forest biomes examined in this study.

Reserve groups and categories are listed in accordance with the reserve classification sanctioned by the National System for Conservation Units (SNUC Law 9,985/2000).

	N	Total area (ha)	Internal deforestation (%)	External deforestation (%)	Mean area (ha)	SD	Smallest (ha)	Largest (ha)	Mean internal deforestation (%)	SD	Mean external deforestation (%)	SD	
Amazonia	
Strictly protected reserves	
Biological reserve	13	4,899,773	2.58	13.57	376,906	330,856	25,068	1,147,322	3.53	7.41	11.32	14.31	
Ecological station	17	10,081,020	0.70	5.62	593,001	1,189,625	126	4,203,563	5.83	17.45	13.19	21.07	
Park	43	26,034,053	0.86	12.47	605,443	792,182	1,192	3,830,538	6.17	13.01	19.33	25.39	
Natural monument	0												
Wildlife refuge	1	6,369	22.20	78.10					22.20		78.10		
Subtotal	74	41,021,215	1.02	11.39	554,341	848,795			5.84	13.51	17.31	23.98	
Sustainable use reserves	
National/state forest	55	27,951,887	1.34	11.33	508,216	699,233	434	3,604,057	6.16	12.53	17.28	21.81	
Sustainable development reserve	19	10,877,457	0.60	0.04	572,498	687,800	22,461	2,421,927	4.92	10.41	11.28	21.82	
Extractive reserve	67	13,605,065	3.07	17.55	203,061	268,757	476	1,289,379	7.46	10.96	30.45	30.55	
Environmental protection area	29	17,833,933	21.37	37.57	614,963	976,274	149	4,447,238	36.54	27.55	48.41	28.29	
Area of relevant ecological interest	4	44,586	4.60	32.04	11,146	9,375	2,574	25,654	12.07	10.70	28.18	18.45	
Natural heritage private reserve	3	798	0.97	28.26	508,216	699,233	9	487	6.16	12.53	17.28	21.81	
Subtotal	177	70,313,726	6.44	16.72	397,253	649,164			11.88	19.28	27.16	28.86	
Total	251	111,334,941	12.11	53.09	443,566	717,431			10.10	17.99	24.25	27.87	
Atlantic forest	
Strictly protected reserves	
Biological reserve	23	202,608	9.54	63.37	8,809	11,592	563	50,873	8.91	8.60	66.12	21.40	
Ecological station	31	150,315	6.89	67.05	4,849	14,028	10	79,528	14.26	18.75	59.09	30.42	
Park	133	2,011,164	8.45	60.49	15,122	33,992	2	303,280	22.54	26.88	60.83	25.74	
Natural monument	10	36,423	37.78	78.50	3,642	5,504	54	17,444	49.54	26.05	71.81	14.71	
Wildlife refuge	9	66,192	29.01	60.07	7,355	8,075	80	23,328	43.81	28.80	67.22	23.17	
Subtotal	206	2,466,701	9.45	62.16	11,974	28,538			22.01	26.13	61.97	25.73	
Sustainable use reserves	
National/state forest	26	36,876	53.70	70.82	1,418	1,650	89	5,385	44.36	37.04	66.27	25.56	
Sustainable development reserve	6	13,415	31.63	37.60	2,236	1,812	665	5,822	28.12	19.04	32.11	21.12	
Extractive reserve	11	71,371	39.53	71.04	6,488	9,643	345	32,755	25.96	24.68	57.73	30.76	
Environmental protection area	116	6,311,474	58.85	70.21	54,409	107,733	18	827,974	54.15	28.26	66.52	22.34	
Area of relevant ecological interest	12	15,473	24.42	66.26	1,289	1,926	82	5,762	36.62	34.88	59.37	29.93	
Natural heritage private reserve	160	39,744	24.21	61.74	248	805	2	7,941	23.04	26.03	59.71	22.60	
Subtotal	331	6,488,352	58.28	67.96	19,602	68,748			36.30	31.50	62.03	23.95	
Total	537	8,955,053	44.49	65.73	16,676	56,916			33.91	30.16	62.01	24.65	

Figure 2 Human population density (HPD) within and outside Amazonian and Atlantic Forest conservation units under different classes of sustainable use reserves (SUR) and strictly protected reserves (SPR).

Diagonal dashed red lines represent equality in HPD within and outside reserves, so that all circles above the line indicate reserves for which HPD outside was greater than that of inside reserves.

Human-modified areas within SURs were usually proportionately much larger than within SPRs (Table 2). In fact, although vegetation conversion rates inside reserve boundaries scaled to conversion rates in the surrounding buffer areas of the same reserves in both biomes (p < 0.001), SPRs comprised a more effective deterrence against deforestation than SURs, particularly for Atlantic Forest reserves for which major class of reserve management was a significant predictor of conversion rates (p < 0.001; Fig. S4). Reserve age (year of decree) had an overall positive effect on conversion rates for Atlantic Forest reserves (p = 0.028) but not for Amazonian reserves (p = 0.717), once reserve management class and conversion rates in external buffers were controlled for. Overall, 45.2% of the aggregate Atlantic Forest reserve area across all categories had been converted to other land-uses (mean = 33.9 ± 30.2%), and therefore fared far worse than Amazonian forest reserves, which had lost only 12.1% (mean = 10.1 ± 18.0%) of their total forest area.

Deforestation rates declined in increasingly larger reserves, which protect larger and more ecologically viable forest areas from structural alterations in surrounding landscapes (Fig. 3; Table 3). Among all reserve categories, Environmental Protection Areas (EPAs) and Wildlife Refuges experienced the highest levels of forest loss, whereas Parks and Reserves performed much better. This is expected since these strictly protected reserve categories aim to conserve natural ecosystems under the public domain, while EPAs are predominantly comprised of private lands under special regulations, often encompassing agricultural and even urban areas. Accordingly, generalized linear models showed that strictly protected reserves were far more effective than sustainable use reserves in deterring vegetation conversion once other variables were taken into account (Fig. S4).

Figure 3 Proportion of forest reserves converted to other land-uses as a function of reserve size for both sustainable use (SUR) and strictly protected reserves (SPR) across the Brazilian Amazon and the Brazilian Atlantic Forest.

Note that larger reserves tend to be less degraded in all cases, except for SURs in the Atlantic Forest.

Table 3 Parameters estimated by the generalized linear models for Conservation Units in either the Amazon or the Atlantic Forest.

Levels of significance are indicated between brackets only for significant variablesa in each model (in bold).

	Amazon	Atlantic Rainforest	Both biomes	
N	252	539	788	
R2	0.616	0.252	0.339	
AIC	–139.67	418.94		
	t ratio	
lag_yrs	0.73	–0.26	1.65	
loghpd_uc	4.05	5.77	6.13	
	<0.0001	<0.0001	<0.0001	
loghpd_buf	1.15	–3.47	–0.82	
		0.0006		
log_uc_dryarea	–3.89	0.94	–3.79	
	0.0001		0.0002	
logpib	1.71	1.32	–0.57	
categ2[EPA]	4.02	3.29	6.28	
	<0.0001	0.0011	<0.0001	
categ2[AREI]	0.35	0.55	0.33	
categ2[FOREST]	1.07	2.45	–0.79	
		0.0148		
categ2[PARK]	0.25	–2.98	–2.92	
		0.003	0.0036	
categ2[REFUGE]	0.22	2.53	3.93	
		0.0117	<0.0001	
categ2[RESERVE]	–0.26	–4.24	–4.77	
		<0.0001	<0.0001	
categ2[NHPR]	–3.12	–3.13	–3.84	
	0.002	0.0018	0.0001	
categ2[SUST]				
Notes.

a AIC, Akaike Information Criterion; lag_yrs, time since conservation unit established (years); loghpd_uc, logarithm of human population density estimated within the conservation unit; loghpd_buf, logarithm of human population density estimated in area surrounding the conservation unit; log_uc_dryarea, logarithm of the dry area of the conservation unit; logpib, logarithm of the weighted mean GDP of the area covered by the conservation unit; Categ2[EPA, etc.], analysis categories for the conservation units.

When we combined all reserve categories across both biomes, human population density (HPD) and human development index (HDI) were the only significant socio-economic variables explaining the degree to which reserves had been degraded (Figs. 4, 5 and Fig. S4). As such, reserves were more degraded in more densely settled areas, but particularly in more developed counties. HPD also consistently declined in increasingly larger reserves, except for sustainable use Atlantic Forest reserves where this relationship was not significant (Fig. 6).

Figure 4 Human population density (HPD) gradient at the scale of sub-municipal census districts across the entire Brazilian Amazon and the Brazilian Atlantic Forest, overlapping the boundaries of the 788 forest reserves examined in this study.

Geographic variation in HPD is expressed across a color gradient including eight orders of magnitude.

Figure 5 Forest reserve conversion ratios as a function of Human population density (HPD) ratios within sustainable use (SUR) and strictly protected reserves (SPR) in the Brazilian Amazon and the Brazilian Atlantic Forest.

Ratios are expressed as the log-transformed (log10 x + 0.1) values for forest conversion (%) and HPD estimates for areas within external buffers outside reserves divided by those within reserve boundaries (Outside:Inside). Dashed horizontal and vertical lines represent equal values within and outside reserves. Rates of forest conversion tend to be higher in reserves exhibiting greater HPD.

Figure 6 Relationship between HPD at a landscape-scale (based on the area-weighed HPD estimate including both the reserve polygon and its surrounding 10-km buffer area) and reserve size for sustainable use reserves (red circles) and strictly protected reserves (blue circles) for both the Brazilian Amazon and the Atlantic Forest.

Linear slopes are not significantly different between reserve classes for Amazonia but significantly different for the Atlantic Forest.

Amazonian reserves

Large conservation units dominate the impressive reserve network amassed throughout the Brazilian Amazon since the early 1980s (mean = 443,566 ± 717,431 ha, N = 251). In total, 1,113,349 km2 (26.6%) of this biome is already protected within the boundaries of conservation units under different denominations. However, this protected acreage is heavily biased in terms of total area and number of reserves towards SURs (IUCN category V), with four of the five dominant categories in terms of aggregate acreage being National and State Forests (25.1%), Environmental Protection Areas (16.0%), Extractive Reserves (12.2%), and Sustainable Development Reserves (9.8%) (see Table 2).

Human-induced land cover change was lower inside forest reserves of any denomination compared to their external buffers. We noted an outlier reserve for which forest conversion has been particularly elevated compared to even sustainable use reserves: some 22.2% of the total area of the Metrópole Wildlife Refuge, located within the metropolitan area of Belém, the state capital of Pará, had been deforested. This relatively small reserve (6,369 ha), however, was decreed in 2010 as a conservation unit from former private farmland, so most of this deforestation occurred prior to reserve creation. Most Amazonian protected areas are, however, located in very sparsely populated regions, so it is unsurprising that both SPRs and SURs in this biome are large and exhibit very low rates of degradation (Fig. 5). Moreover, human population density rapidly decreases by four orders of magnitude across the size range of Amazonian reserves, particularly those under strict protection (Fig. 3).

Atlantic Forest reserves

A total of 537 Atlantic Forest reserves were examined, including 133 Parks, 160 NHPRs, and 116 EPAs. In total, these conservation units of different denominations amount to 89,551 km2 (8.0%) of this biome under any degree of protection. Protected areas in the Atlantic Forest tend to be far smaller and surrounded by more degraded land than those in Amazonia (Fig. 7). EPAs comprised the largest category of sustainable use reserves (mean size = 54,409 ± 107,733 ha, N = 116) but had experienced the highest levels of forest conversion (Table 2). Levels of legally permitted human activities and high HPD within EPAs has resulted in a high degree of within-reserve degradation (58.7%). However, large strictly protected Atlantic Forest reserves have been much more effective at inhibiting forest conversion (Fig. 3). As such, reserve size failed to explain the degree to which SURs had been degraded, with many large SURs also exhibiting high proportions of deforestation. The most intact biogeographic subregion of the Atlantic Forest lies within the Serra do Mar montane domain, which still retains some 36.5% of its original vegetation cover. This high-elevation subregion contains the largest strictly protected Atlantic Forest reserves within the densely populated states of São Paulo, Rio de Janeiro and Paraná, which account for 35.5% of the Brazilian population.

In contrast, strictly protected reserves such as biological reserves, ecological stations and parks (IUCN categories Ia, Ib and II, respectively) exhibited much lower deforestation rates, typically well below 10%. Reserve management category was therefore more important than reserve size per se in deterring deforestation across the Atlantic Forest. High levels of degradation within external buffers (mean = 62.0 ± 24.7%) indicate that the heavily settled landscapes surrounding Atlantic Forest reserves have become highly fragmented for both strictly protected and sustainable use reserves (Fig. 8).

Figure 7 Overall size distribution of 251 Amazonian and 537 Atlantic Forest sustainable use (SUR) and strictly protected (SPR) forest reserves (Mean area, Range; Amazonian SURs: 382,142 ha (9—4, 247,778 ha); Amazonian SPRs: 541,570 ha (126—4,196,585 ha); Atlantic Forest SURs: 21,963 ha (2—776,700 ha); Atlantic Forest SPRs: 11,781 ha (1—301,834 ha)).

Figure 8 Proportion of original forest area lost outside reserves (10-km buffer areas) as a function of proportional forest area lost inside the same reserve for both sustainable use (SUR) and strictly protected reserves (SPR) across the Amazon and the Atlantic Forest.

Diagonal dashed red lines represent equality (1:1) ratios. Forest conversion rates are typically above these lines (i.e., lower within than outside the vast majority of reserves), except for Atlantic Forest SURs for which many conversion rates inside reserves were actually higher than those outside.

Discussion

Several studies have considered the effectiveness of protected areas in terms of biodiversity conservation (e.g., Bruner et al., 2001; Nepstad et al., 2006; Coetzee, Gaston & Chown, 2014; Bradshaw, Craigie & Laurance, 2015). These studies have typically examined a small number of PAs at global or continental scales. However, global analyses using small sample sizes per country can mask national or regional trends in the de facto protection and true effectiveness of protected areas (Schmitt et al., 2009). In contrast, we considered anthropogenic conversion of natural vegetation into different forms of land use within 788 strictly protected or sustainable use reserves within the two largest neotropical forest domains. There were clear contrasts between the Amazon and the Atlantic Forest biomes, particularly in terms of the size structure of nature reserves, overall levels of natural habitat degradation, which reflect major differences in regional scale socio-economics and human population density across those two biomes. This in turn results from clear differences in post-colonial trajectories in human occupation, frontier expansion and land use, which paved the way to arguably the greatest polarity in economic development for two major regions within a single tropical country.

The Atlantic Rainforest region was the first in Brazil to be settled by Europeans, following through several economic cycles based on resource extraction, including Pau-Brasil (Caesalpinia echinata) exploitation, before the emergence of sugarcane and coffee agriculture (Joly, Metzger & Tabarelli, 2014). New development and national integration cycles then led to industrialization and urbanization which were largely confined to eastern Brazil, drastically reducing its natural vegetation. Today, 70% of all ∼205 million Brazilians live within the Atlantic Forest domain, including Brazil’s largest metropolitan centers. The pace of forest conversion was faster in the 20th century, leading to a high degree of habitat fragmentation and biodiversity loss (Ribeiro et al., 2009). Forest conversion and fragmentation affect both protected and unprotected areas, reflecting the most heavily settled, most intensely farmed, and wealthiest parts of Brazil (Oliveira & Oliveira, 2011). As a result, the Atlantic Forest has become one of the most threatened tropical biodiversity hotspots worldwide (Myers et al., 2000). In addition, most forest remnants across this biome are smaller than 50 ha and approximately 9% of all remaining natural vegetation cover is protected by conservation units (Ribeiro et al., 2009), but these are disproportionally concentrated at high elevation areas where agricultural opportunity costs are lower (Tabarelli et al., 2010). Even after the enactment of the Atlantic Rainforest Law (Law 11,428/2006), which added further restrictions to forest conservation legislation in Brazil, deforestation rates across the Brazilian Atlantic Forest remain high, averaging 22,384 ha per year (Fundação SOS Mata Atlântica & INPE, 2013). Even strictly protected reserves under human encroachment pressure tend to lose peripheral forest cover (Terra, Dos Santos & Costa, 2014). This is driven both by forest conversion to agriculture and firewood extraction. For example, in northeastern Brazil, up to 76% of all rural households still rely on wood for fuel, consuming 0.96 ton person-1 of tree biomass each year (Specht et al., 2015). As increasingly fewer people comply with this law, conservation units under the public domain become even more important, as shown by the abysmally poor performance of EPAs in deterring agricultural and urban expansion and retaining forest cover. This further emphasizes the critical role played by strictly protected conservation units in this biome despite their timid expansion in total area since the early 1960s (Fig. S3).

As many conservation units are now near urban areas or completely surrounded by farmland, land-use restrictions often come under economic pressure. EPAs (IUCN category V) are the largest and numerically dominant type of conservation unit throughout the Atlantic Forest. This is the only reserve category that, at once, attempts to meet the intractable challenge of protecting biodiversity while controlling human occupation and managing agricultural and urban expansion. However, EPAs provide the lowest degree of de facto protection due to the history of land use, large resident populations, and land tenure conflicts (Viana & Ganem, 2005). Furthermore, EPAs are typically established in areas of high human encroachment pressure, and environmental agencies often justify setting them aside as the best way to prevent further land use change. EPAs are therefore less important in terms of biodiversity conservation, even though they are defined as part of the SNUC network of PAs (Pádua, 2012). However, they deceptively contribute vast additional tracts of land to the Convention on Biological Diversity Aichi Target 11, which requires that each signatory country should allocate at least 17% of its land area by 2020 to terrestrial protected areas (Juffe-Bignoli et al., 2014). Although all Atlantic Forest conservation units currently account for 8.0% of the entire biome area, this remains a remote target even if one disregards the overall poor performance and high forest conversion rate of many forest reserves (∼44.5% in SPRs and ∼58.3% in SURs, see Table 2).

In terms of the size structure of existing reserves, at one extreme a large number of forest reserves have been set aside within private landholdings (NHPRs), mostly within the Atlantic Forest (IUCN Category IV). These typically small reserves tend to be embedded within highly fragmented landscapes. However, conservation restrictions tend to be well enforced within NHPRs under the watchful eyes of protective private landowners, which often perform reasonably well compared to other SURs (∼24.2% of forest conversion). Small forest fragments, however, provide limited conservation services in retaining wide-ranging species (Joppa, Loarie & Pimm, 2008), particularly in the long-term retention of many threatened species that are not necessarily considered in official red lists (Schnell et al., 2013). Many NHPRs and other small reserves also succumb to a double jeopardy whenever surrounding landscapes are heavily settled and subjected to intensive land use (Parks & Harcourt, 2002; Benchimol & Peres, 2013). Yet these conservation units still play an important conservation role, particularly in densely populated parts of Atlantic Forest, where forest remnants are now distributed across over 245,000 often small forest fragments (Ribeiro et al., 2009). These remnants increase landscape connectivity between large fragments, can boost population sizes, and operate as a refuge in case any major disturbances (e.g., wildfires) take place in larger protected areas (DeFries et al., 2005). NHPRs comprise the most ubiquitous and numerically dominant reserve category in the Atlantic Forest, and they continue to proliferate each year, mainly due to land tax incentives. These reserves may not be self-sufficient but greatly complement landscape-scale conservation planning, and their start-up costs are virtually zero for the public treasury because they are privately owned and managed. Moreover, the risk of any intentional deforestation after a private reserve is created is negligible, as even a change in property ownership does not entitle any new landowner to any changes in land use. NHPRs already exceed forest reserves under public jurisdiction in both total area and numbers, and will increasingly play an important role in balancing biodiversity conservation, provision of ecosystem services and human welfare particularly in heavily human-modified landscapes (Melo et al., 2013).

The Brazilian Amazon, on the other hand, has experienced unprecedented deforestation rates, but unlike the Atlantic Forest, this recent process of frontier expansion largely took place in the last four decades. New and extensive roads have been built, including the BR-230 (Transamazon Highway), BR-319 (Manaus—Porto Velho Highway), BR-163 (Cuiabá—Santarem Highway), and the BR-156 Highway from Amapá to French Guiana. Further construction of new roads, major hydroelectric dams, and oil and gas pipelines are part of an ambitious set of investments to enhance regional infrastructure and open up new development frontiers (Soares-Filho et al., 2005). These investments, particularly new roads, stimulate agricultural frontier expansion, and pave the way to human migration and access to hitherto unexploited timber resources (Schneider & Peres, 2015). The growing network of paved and unpaved roads also facilitate ‘land grabbing’ of public lands, timber extraction and wildfires both within and outside poorly implemented and rarely enforced protected areas (Souza, Roberts & Cochrane, 2005). With rapid economic changes, the region has seen an economic transition from resource extractivism to industrialization, with mineral exploitation and commodity production from agribusiness such as cattle and soybean gaining ground (Soares-Filho et al., 2005) and driving deforestation (Barona et al., 2010). Public policies for credit, subsidies, land occupation and resettlement of southern Brazilian farmers have also encouraged deforestation (Fearnside, 2005; Schneider & Peres, 2015). In line with these changes, Brazilian Amazonia exhibits the highest urban growth rate (Oliveira & Oliveira, 2011). This means that each economic activity contributes individually or collectively to current or future deforestation rates (Fearnside, 2005). Under the most pessimistic deforestation forecasts, forest losses by 2050 may exceed 45% of the Brazilian Amazon (Soares-Filho et al., 2005). However, data from the Brazilian Space Agency (INPE, 2015) indicate an average annual reduction of 13.5% in deforestation rates over the last decade lending room for some optimism.

The Amazon is seen worldwide as one of the last remaining natural capital frontiers on Earth (Becker, 2005), but faces high expectations within Brazil in terms of valuing standing forests. Despite international instruments proposed to slow down deforestation, such as the Reduced Emissions from Deforestation and Degradation (REDD+) program and other carbon emission exchange mechanisms, these expectations have amounted to very little in practice and should be seen as complementary to on the ground conservation efforts. Consolidating the network of Amazonian protected areas, regardless of their legal categories, provides an effective contribution to biodiversity conservation and all associated ecological processes, mainly by sequestering public lands that would otherwise be widely available to further land-use change including deforestation.

Although the sheer size of many Amazonian forest reserves is decisive in slowing down deforestation, there are major differences between reserve categories in terms of forest conservation performance. For example, deforestation rates within EPAs was nine-fold greater than those recorded in SPRs (Table 2), and this is facilitated by no legal restrictions on agricultural land use, including slash-and-burn subsistence agriculture, which reduces secondary-forest resilience and crop productivity (Jakovac et al., 2015), therefore demanding ever more forest conversion to support the livelihoods of a growing population. All other categories of sustainable use reserves, however, exhibited forest conversion rates lower than 5% at least for now, even when their surrounding areas had already been deforested. Deterring these encroachment pressures, however, will require sustained government action, including effective vigilance, law enforcement and a good working relationship with local communities, particularly in legally occupied forest reserves.

The official count of 1,602 continental conservation units in Brazil, which currently represent 17.2% of the Brazilian territory, indicates that current international targets in safeguarding native biodiversity have already been reached (MMA, 2012). However, when we assess the aggregate existing conservation acreage by management category, official assertions on the degree to which Brazilian ecosystems are effectively protected become greatly overestimated. Since 2003, we have witnessed an increase of 47.3% in the number of protected areas established, especially sustainable use reserves in the Amazon. Sustainable use reserves, of often questionable long-term future, now far exceed strictly protected reserves both in Brazil (Peres, 2011) and worldwide (Jenkins & Joppa, 2009; Schmitt et al., 2009), and have proved to be less effective than strictly protected areas in the Brazilian Cerrado (Paiva, Brites & Machado, 2015). In addition, the most permissive reserve category (EPAs), whose land-use restrictions are effectively negligible, account for some 19% of the entire extent of protected areas across the Amazon and the Atlantic Forest. As a result, 58.9% and 21.4% of the total area of EPAs in the Atlantic Forest and Amazonia, respectively, has already been deforested.

One of the factors contributing to the expansion of multiple-use tropical forest reserves is the high financial cost of establishing strictly protected conservation units, especially in the Brazilian Atlantic Forest, as this involves expropriating private land and removing the resident population as required by law. In Brazilian Amazonia, only 24% of all lands are privately owned and much of the region remains sparsely populated. This facilitates the creation of large forest reserves and indigenous territories, which cover an additional 21.7% of the entire Legal Amazon territory (Ricardo, 2011). Furthermore, sustainable use reserves are politically more viable and more socially acceptable than strictly protected reserves, particularly in densely populated areas (Nelson & Chomitz, 2011). In many cases, when a protected area is set aside, government mandated implementation actions that ensure protection may not necessarily follow, such as developing the appropriate infrastructure, removing squatters and effective monitoring. Therefore, this often disconcerting lack of reserve implementation may justify official downgrading from strictly protected to sustainable use reserves or downsizing reserve boundaries to exclude heavily degraded areas (Bernard, Penna & Araújo, 2014; Marques & Peres, 2015).

There are also additional factors that exert social and economic pressure at a regional scale, which may provoke changes in federal environmental legislation affecting protected areas in both biomes. One risk factor is the political pressure for formal alterations to the legislative acts that create a protected areas area in the first place. Currently, there is a growing number of bills circulating in the Brazilian Congress with the aim of altering environmental legislation, directly affecting the status of conservation units and other protected areas in Brazil. Legal initiatives to reduce, cancel or otherwise alter the protection status of 27 federal reserves nationwide are currently under appreciation by Congress (Marques & Peres, 2015). These alterations translate into further protected area losses, compromising national conservation targets to meet binding agreements under the Convention for Biological Diversity.

This study therefore shows that the mere presence of conservation units established on paper inhibits deforestation even under scenarios of dismal implementation investments after 5 years or more of reserve creation. However, in many regions, reserve size is less important to future reserve performance than the management category. For instance, with the exception of private reserves, the forest conservation performance of sustainable use conservation units is very poor in heavily settled post-frontier regions, such as the Atlantic Forest and in increasingly degraded subregions of Amazonia where protected areas now contain all remaining forest cover (Pedlowski et al., 2005). In the Atlantic Forest, strictly protected reserves, preferably under the public domain, continue to be essential in retaining relatively intact natural ecosystems. In contrast, human population pressure is much lower in most of the Amazon, so that physically demarcated reserves, be they sustainable use or strictly protected, are very efficient for now in maintaining relatively intact forest cover, with increasingly larger reserves performing well under different landscape contexts of external encroachment. This picture may change, however, as large infrastructure projects pave the way to agricultural expansion and burgeoning human populations inflated by economic migrants. Investments in protected area defense and enforcement of reserve management plans will therefore need to scale to growing external pressure, or else we risk undoing much of the huge gains in conservation acreage over the last four decades that has earned Brazil a unique contribution in global scale environmental targets and protected area expansion.

Supplemental Information

Figure S1 Spatial overlaps removal

Examples of GIS treatment of spatial overlaps that are often found between forest reserves in Brazil: (A) a state-level reserve created in 1984 (dotted area) partially overlaps a federal reserve created in 1982 (hatched area), whereas the reserve buffer zone (shaded area) overlaps another reserve; (B) overlaps are manually removed, whereby the boundaries of the older reserve prevails; (C) a strictly protected reserve was created in 2001, partially overlapping a sustainable use reserve created in 1997; and (D) the overlap was then removed to maintain the more recent (and more restrictive) reserve in the sample.

Click here for additional data file.

Figure S2 Spatial distribution of the sample

Distribution of all 788 reserves examined in this study across the Brazilian Amazon and the Brazilian Atlantic Forest biomes. Circle sizes are proportional to the log-transformed area (log10 x, hectares) of each reserve. Color gradient of reserve centroids indicate the degree to which their natural forest cover had been converted into other land uses (from blue to red indicating least to most degraded).

Click here for additional data file.

Figure S3 Protected area increase over time

Temporal growth in total reserve acreage in Amazonia and the Atlantic Forest biomes, broken down by major reserve types (SUR, sustainable use reserves; SPR, strictly protected reserves) (data sourced from the CNUC database).

Click here for additional data file.

Figure S4 GLM coefficient estimates

Coefficient estimates (±95% confidence intervals) showing the magnitude and direction of effect sizes of different reserve size and socioeconomic predictors of the degree to which 788 forest reserves had been converted to other land uses across the Brazilian Amazon and the Brazilian Atlantic Forest. Only variable retained in the most parsimonious final models are shown, including HPD (Human Population Density), HDI (Human Development Index) and major class of reserve category (SURs and SPRs). For a description of predictor variables, see Methods.

Click here for additional data file.

The two premier Brazilian legislative bodies, the Câmara Legislativa do Distrito Federal and Câmara dos Deputados made this analysis possible by approving study leaves for ABM and MS.

Additional Information and Declarations

Competing Interests

Author Contributions

Data Availability

The authors declare there are no competing interests.

Ana Alice B. de Marques, Mauricio Schneider and Carlos A. Peres conceived and designed the experiments, performed the experiments, analyzed the data, wrote the paper, prepared figures and/or tables, reviewed drafts of the paper.

The following information was supplied regarding data availability:

All data used in this paper came from the latest official sources in the executive branch of the Brazilian Federal government, covering political boundaries, protected area polygons, vegetation and socioeconomic variables: www.ibge.gov.br, www.inpe.br, www.pnud.org.br, www.icmbio.gov.br, mapas.mma.gov.br/i3geo, siscom.ibama.gov.br, and mapas.mma.gov.br/i3geo.

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
