# Peer review of "Human population and socioeconomic modulators of conservation performance in 788 Amazonian and Atlantic Forest reserves"

_PeerJ, doi:10.7717/peerj.2206_

## Round 0.1 · original submission · Major Revisions

Dear authors,

I have received the review reports on your manuscript, "Nonlinear temperature effects on multifractal complexity of metabolic rate", which you submitted to PeerJ. Based on the advice received, your manuscript could be reconsidered for publication should you be prepared to incorporate major revisions. Especially, the following review comments should be considered:

1 -In FIgure 4, the colour of polygons of strictly protected areas are not easily readable, hope to be changed.

2- A short comparison between the results in Amazonian reserves and Atlantic forest reserves should be provided and some discussions should given to the impacts of conservation performance corresponding to different levels of human activity. It is better to use a table to indicate the differences and characteristics on the two regions.

3- In the Methods, it is very helpful and important for the authors to construct a logic framework to explain and show the relations and processes among the forest reserves, biophysical and socioeconomic variables, the different scales of time and space, and policy. That is, the authors should think about methodology on forest reserves issues and integrate these different relations and variables into a whole thing. Maybe a figure can show it clearly.

Reviewer 1 ·

Basic reporting

(1) Line 47: The authors provide one reference to indicate that there are few studies in examining the concomitant roles of both size and management category. The referred paper was published in 2008. I think more updated references are needed.
(2) Also, to supporting your experimental design, reviews on what indicators are used to study the reserve conservation performance in existing literature are needed in introduction part. You need to answer the question that why human population density and reserve management category are essential to the reservation level of PAs. As you indicates in Line 49, the category of human activities may also important.
(3)Line 210 "an" should be "and"

Experimental design

(1)Line 136 and Table 1: The authors suggest that IUCN IV are belong to sustainable use reserves in SNUC legislation in line 136. But in table 1, IUCN IV are grouped into strictly protected reserves. Is there standard for your grouping? Please provide your principles or references.
(2)Line 206: What is the data source for HDI?

Validity of the findings

(1)Line 265: The comparison among reserve categories should be provided in either table or figure format. If neither, at least the cumulative reservation rate should be provided here. And what about the impact of other categories?
(2)Line 265: In your experimental design, eight categories are provided in table 1. However, in your findings, the original categories such as wildlife refuges are used. Please unify the name you used. This question is related to the variable you use as "legal reserve category”. Does it mean the IUCN categories? If so, why you need to group 8 classes at beginning?
(3)Line 274: Since the title of this paper is "human population and ...", the result of human population and human development index should be discussed more.
(4)Line 676: Shown in figure 4, the HPD may be highly related to the reserve class (SPRs or SURs). If so, it may means that there are not only the inner-group differences in HPD but also significant inter-group differences in HPD. Therefore, the HPD may be highly related reserve class.
(5)Line 274, the impact of HDI should also be provided.

Additional comments

The paper studied the impact of human population and reservation category to habitat conversion rates. The theme, purpose and method are clear. The paper is well organized and written in an appropriate style. However, there are some details needs to be revised.

(1) In your abstract, there is a sentence saying that "our results show that PAs surrounded by heavily settled agro-pastoral landscapes face much greater challenges in retaining their natural vegetation". Which variable measures the "agro-pastoral landscapes"?
(2) In the discussions, the authors should emphasize the contribution of this paper by comparing their findings with existing research and try to discuss any inconsistencies which can be future research directions.

·

Basic reporting

The submission adhere all PeerJ policies and have the components of the basic reporting. But I suggest two things:
1. State your hypothesis explicitly toward the end of the introduction.
2. There are some awkward phrases and errors in word choice, for example “Amazonia biome”, the correct form is “Amazon biome”.

Experimental design

The Experimental Design is described with sufficient information to be reproducible by another investigator, but the dates that the maps of vegetation were used are incompatible. The deforestation map of Atlantic Forest biome was 2008 and for the Amazon biome was 2011. If the work is to compare these two biomes mainly with the socioeconomic factors and population density it is necessary to use the same date. I recognize that latest map for Atlantic Forest biome is 2008, however to the Amazon biome we have 2008 from two data sources; PRODES and TERRACLASS. Furthermore, the vegetation in Amazon biome was represented by PRODES at a scale of 1:250.000, instead of TERRACLASS map was created at a 1:100.000 scale. What is the main reason for not using TERRACLASS Amazon? Once the mapping scale of Atlantic Forest (PMDB) is 1:50.000, if used the mapping of TERRACLASS Amazon the work will be more detailed, because the scale can be 1:100.000.
And the last thing is the Figure S3, in which explain about the temporal growth of reserve but did not write the author reference, neither in the body text.

Validity of the findings

The conclusions were appropriately stated.

Additional comments

I would like to end by reinforcing change - maps for Amazon biome; instead of PRODES 2011 by TERRACLASS Amazon 2008, as well as the final scale from 1:250.000 to 1:100.000.

---

## Round 0.2 · Minor Revisions

I have received the comments on your revised manuscript from two reviewers. One of them recommended your manuscript to be published as its current status, but another advised you to revise manuscript again.

After reading your revised manuscript, I found that your revised version has indeed been improved. However, I think you should revise your manuscript according to the review comment, or give some reasonable explanation addressing the review comment.

The review comments for the major revision on your manuscript are as follows:

I would like to end by reinforcing change - maps for Amazon biome; instead of PRODES 2011 by PRODES 2008. Certainly the article will waste quality comparing two too different areas with two different dates. If you don't have data for biome Amazon in 2008, you can do with data you have, because we need researches like yours. However, if has the data (PRODES 2008) there is no good reason to not to use them.

Reviewer 1 ·

Basic reporting

No Comments

Experimental design

No Comments

Validity of the findings

No Comments

Additional comments

The reivision has met the standard for publication.

·

Basic reporting

No comments

Experimental design

The authors said that "TERRACLASS deforestation data is actually the same as the PRODES deforestation data for 2008", which it is near-truth. The difference is that TERRACLASS qualifies deforestation (which land use is in the place of vegetation) that were present in PRODES. Therefore, in my vision the article can be enriched using TERRACLASS, but I leave this with authors' judgment.

Finally have been mentioned about the years 2008 (biome Atlantic) and 2011 (biome Amazon) in the other review, and I would reiterate the importance of compare two same dates, because the socioeconomic factors, population density, per capita wealth and other facts were very different in 2008 and 2011.

Validity of the findings

No comments

Additional comments

I would like to end by reinforcing change - maps for Amazon biome; instead of PRODES 2011 by PRODES 2008. Certainly the article will waste quality comparing two too different areas with two different dates. If you don't have data for biome Amazon in 2008, you can do with data you have, because we need researches like yours. However, if has the data (PRODES 2008) there is no good reason to not to use them.

---

## Round 0.3 · accepted · Accept

After reading the second version of your revised manuscript, I decided to accept your article for publication in PeerJ.